# Therapeutic Aqueous Humor Concentrations of Latanoprost Attained in Rats by Administration in a Very-High-Molecular-Weight Hyaluronic Acid Eye Drop

**DOI:** 10.3390/pharmaceutics16040523

**Published:** 2024-04-09

**Authors:** Kazunari Higa, Reona Kimoto, Takashi Kojima, Murat Dogru, Wolfgang G. K. Müller-Lierheim, Jun Shimazaki

**Affiliations:** 1Cornea Center Eye Bank, Tokyo Dental College Ichikawa General Hospital, 5-11-13 Sugano, Ichikawa 272-8513, Japan; bookmoto0812@gmail.com (R.K.); meishano1@gmail.com (J.S.); 2Department of Ophthalmology, Keio University School of Medicine, 35 Shinanomachi, Tokyo 160-8582, Japan; tkojkoj@mac.com (T.K.); muratodooru2005@yahoo.co.jp (M.D.); 3i.com Medical GmbH, 81241 Munich, Germany; ml@coronis.net; 4Department of Ophthalmology, Tokyo Dental College Ichikawa General Hospital, 5-11-13 Sugano, Ichikawa 272-8513, Japan

**Keywords:** dry eye, anti-glaucoma eye drops, hyaluronic acid eye drops, aqueous humor

## Abstract

The temporal change in concentration of a novel medicine, Latanoprost (LP), was evaluated in the aqueous humor of rats (6–8-week-old Jcl:Wister rats) when delivered in a very-high-molecular-weight hyaluronic acid (vHiHA) eye drop. Animals were randomly assigned to three treatment groups (LP + vHiHA (LPvHiHA), commercial LP (cLP), and diluted LP (dLP)) and after instilling the eye drops, the aqueous humor (AH) was collected at 0.5, 1, 2, 4, and 6 h to measure the LP concentration using an enzyme-linked immunosorbent assay (ELISA). Although the LP concentration in the LPvHiHA eye drop formulation was 3.57 times lower than in the commercial eye drops used (cLP), the LP concentration in the AH following LPvHiHA administration reached a value close to that of cLP. The cLP was diluted to the same concentration of LP as in the LPvHiHA eye drops for the dLP group, but the LP concentration in the AH of these animals was lower than that of the LPvHiHA rats at all time points. The higher LP concentration in the AH of the LPvHiHA rats suggests that vHiHA may aid the transport of LP across the ocular surface epithelium.

## 1. Introduction

Prostaglandin analogues (PGAs) are used as first-line anti-glaucoma agents and they are known to be very effective in decreasing intraocular pressure (IOP) [1,2]. However, long-term treatment with topical anti-glaucoma medications can have adverse effects on the ocular surface, potentially inducing conjunctival hyperemia, keratitis, follicular conjunctivitis, iris and periocular skin pigmentation, eyelash growth, or herpes reactivation [3,4]. Most of the eye drops used to deliver Latanoprost (LP), a PGA used to reduce elevated IOP, contain preservatives like benzalkonium chloride (BAC), and these may cause ocular surface epithelial disorders [5]. These problems may be alleviated by using preservative-free LP eye drops, although even so, ocular surface damage may develop due to inflammation-induced dry eye [6,7,8]. Moreover, all currently available commercial eye drops used to deliver LP contain penetration enhancers that compromise the ocular surface, such as surfactants and/or ethylene diamine tetraacetic acid (EDTA), the long-term application of which may also be detrimental to the ocular surface [9].

Hyaluronic acid (HA) is a major carbohydrate component of the extracellular matrix (ECM) and a high-molecular-weight glycosaminoglycan that is widely distributed throughout the body. High-molecular-weight HA (HiHA: 1.5–1.7 MDa, intrinsic viscosity 2.5–<2.9 m^3^/kg) and very HiHA (vHiHA: ≥1.8 MDa, intrinsic viscosity ≥ 2.9 m^3^/kg) [10,11] have anti-inflammatory and wound-healing effects that are associated with their anti-angiogenic and immunosuppressive properties, and they are widely used in the treatment of dry eye [12,13,14,15,16,17,18,19]. These studies suggest that when an anti-glaucoma medication like LP is combined with a vHiHA-containing vehicle, IOP may be decreased with only minor effects on the ocular surface.

We previously demonstrated that the delivery of LP in an eye drop with vHiHA (LPvHiHA) had favorable effects on ocular surface staining and tear functions [20]. Indeed, more recent data indicated that the ocular surface of wild-type (WT) mice receiving LPvHiHA eye drops over 7 days was in better health than that of mice receiving a commercial LP (cLP) eye drop. This improvement was witnessed through the significantly lower corneal vital staining scores, stronger ZO-1 mRNA expression, and higher conjunctival goblet cell densities [20]. In fact, the mice administered LPvHiHA also had significantly less ocular surface–tear film inflammation, as evident through the significantly fewer conjunctival inflammatory CD45+ stained cells and lower tear IL-6, as well as lower IL-1b levels [20]. However, whether these favorable effects were due to better penetration into the aqueous humor (AH) remain to be assessed.

Accordingly, the purpose of this study was to test the hypothesis that an eye drop formulation containing LP in a vehicle that only consists of substances naturally present at the ocular surface may achieve a therapeutic concentration of LP free acid (LPFA) in the AH. The temporal changes to the LPFA concentrations in the AH were measured when LPvHiHA eye drops were applied, and compared to those achieved with a cLP formulation or a diluted version of the same (dLP).

## 2. Materials and Methods

### 2.1. Animals

All experimental procedures and protocols were approved by the Animal Care and Use Committee at the Tokyo Dental College, and they conformed to the National Institutes of Health Guide for the Care and Use of Laboratory Animals. Male (*n* = 21) and female (*n* = 19) Jcl:Wister rats (6–8 weeks of age) were purchased from CLEA Japan, Inc. (Tokyo, Japan), and 5 animals of each sex were assigned to each of the three groups (LPvHiHA, cLP or dLP), with 6 male and 4 female rats used as a quality control. The rats were anesthetized by intraperitoneal administration of a mixture of medetomidine hydrochloride (0.5 mg/kg, Domitor: Meiji Seika Kaisha, Tokyo, Japan), midazolam hydrochloride (2.0 mg/kg, Dormicaum: Astellas Pharma, Tokyo, Japan) and butorphenol (0.5 mg/kg, Vetorphale: Meiji Seika Kaisha), while ocular anesthesia was achieved with oxybuprocaine hydrochloride eye drops (0.4%, Benoxil: Santen Pharmaceutical Co., Ltd., Osaka, Japan).

### 2.2. Preparation of Eye Drops

LPvHiHA eye drops (14 μg/mL LP with vHiHA—a new ophthalmic solution) were provided by i.com medical GmbH (Munich, Germany). These eye drops were derived from a sterile bulk solution of the commercially produced Comfort Shield^®^ eye drops (i.com medical, Munich, Germany), which contained 1.5 g/L very-high-molecular-weight sodium hyaluronate (intrinsic viscosity 2.9 m³/kg), 1.20 mmol/L sodium phosphate buffer and 8.0 g/L sodium chloride dissolved in purified water. This solution was used as the vehicle to dissolve LP at 14 μg/mL (Yonsung Fine Chemicals, Hwaseong-si, Gyeonggi-do, Republic of Korea). The resulting solution was filtered through sterile 0.22 µm pore size Sartorius stedim Biotech Minisart^®^ filter cartridges (Sartorius, Göttingen, Germany). The cLP eye drops were purchased from Rhoto Nitten Co., Ltd. (50 μg/mL commercial LP: Nagoya, Japan), while the dLP eye drops were prepared from 50 μg/mL cLP diluted to 14 μg/mL with saline. The complete list of ingredients and the concentration of LP in the eye drops are indicated in Table 1.

### 2.3. Eye Drop Instillation and Collection of the Aqueous Humor (AH)

The eye drops (15 μL) were instilled into the eyes of each group of male and female rats (*n* = 5), and the quality control rats (males *n* = 6 and females *n* = 4), using a 20 μL pipette. AH (5 μL) was collected with a 32 G needle attached to the cut tip of a 20 μL pipette at 0.5, 1, 2, 4, and 6 h after instillation of the eye drops. As a quality control, AH was also collected from the contralateral eye. The AH samples collected were stored at −80 °C until they were analyzed.

### 2.4. LPFA Measurement in the AH

The LPFA concentration in the AH was measured with an enzyme-linked immunosorbent assay (ELISA: Latanoprost ELISA kit, Cayman Chemical, Ann Arbor, MI, USA) based on competition between the LP and the LP tracer (a LP acetylcholinesterase conjugate) for a limited number of LP-specific antiserum binding sites. The measurements and calculations in this assay were performed according to the protocol provided by the manufacturer. LP is a prodrug that is converted to the active LPFA form on its way through the cornea. The AH samples collected were diluted 500–15,000 times to measure standard LPFA concentrations between 3.9 and 500 pg/mL. The sample plate concentration was measured at a wavelength of 405 nm in a microplate reader (Model 550: BioRad, Hercules, CA, USA).

### 2.5. Osmolarity of the LP Eye Drops

The osmolarity of each LP eye drop was measured with an automatic freezing-point osmometer (Hermann Roebling MESSTECHNIK, Berlin, Germany). Osmolarity was measured in triplicate and the data were shown as an average. The osmolarity of the LP eye drops is indicated in Table 1.

### 2.6. Statistical Analysis

Statistical comparison of the ELISA LPFA concentrations was performed with a non-paired Student’s *t*-test, Welch’s *t*-test or a Mann–Whitney U test, using the Shapiro–Wilk test to determine a normal distribution and the F test to determine variance. These studies were performed with EZR software 4.2.3 (Saitama Medical Center, Jichi Medical University, Saitama, Japan), a graphical user interface for R (The R Foundation for Statistical Computing, Vienna, Austria). More precisely, it is a modified version of the R commander designed to add statistical functions frequently used in biostatistics. We also performed statistical analysis with a repeated measurement ANOVA to assess temporal changes in AH concentrations across the groups. *p* < 0.05 was considered statistically significant.

## 3. Results

In this study, the LPFA concentrations in the AH measured by ELISA ranged from 16 to 6341 ng/mL (Table 2), while after 6 h all the LPFA concentrations in the AH were below 400 ng/mL (Table 2).

There were only significant differences in the LPFA concentrations between males and females in each eye drop group at two time points, at 0.5 h after cLP instillation and 6 h after dLP instillation (Figure 1, asterisks). Although no differences due to sexual dimorphism were detected in this study, there was a tendency towards slightly higher LPFA concentrations in the AH at many time points (Figure 1).

The LPFA concentrations produced by LPvHiHA tended to be higher than those associated with dLP instillation (with the same LP concentration as LPvHiHA) in females but not in males (Figure 2A,B). Moreover, in the combined data of both sexes, significant differences were observed between the LPFA concentrations associated with LPvHiHA and dLP administration at 0.5 and 2 h (Figure 2C, double asterisks). No significant differences were found between the three groups in a repeated measurements ANOVA analysis of the data from males (Figure 2A), yet significant differences were found between the three groups in a repeated measurements ANOVA analysis of the data from females, and from both sexes combined (Figure 2B,C).

To investigate these differences in more detail, we assessed the LPFA concentrations in the AH relative to the ratios of the LP concentrations in the eye drops (cLP/LPvHiHA = 3.57, cLP/dLP = 3.57 and LPvHiHA/dLP = 1.0), using the average of the 5 measurements in Figure 2 (at 1.5, 1, 2, 4 and 6 h: Figure 3). In females, or when assessing the combined data of both sexes, the cLP/dLP concentration ratio between the eye drops (3.57) did not affect the AH concentration (2.74 ± 0.20, 5.73 ± 5.09) simply as a result of dilution (cLP/dLP). However, the AH concentrations for the cLP/LPvHiHA ratios (1.93 ± 0.84, 2.11 ± 1.07) were lower than that of the eye drop concentrations (3.57). Conversely, the LPvHiHA/dLP AH concentrations (2.72 ± 1.19, 1.64 ± 0.42) were higher than the ratio of the eye drop concentration (1.0, *p* < 0.05, *n* = 5: Figure 3B,C). As a quality control, the LPFA concentration in the AH of the contralateral untreated eye was measured and compared to that instilled with LPvHiHA or cLP (Appendix A). There was no change in the LPFA concentration in the AH of the untreated contralateral eyes, with a concentration below 40 in the male or female rats (Appendix A).

To estimate the effect of the LP eye drops on the ocular surface, we also measured the osmolarity of each eye drop. The osmolarity of LPvHiHA was 255 mOSM, cLP was 276 mOSM, and dLP was 302 mOSM. Thus, the LPvHiHA eye drop had the lowest osmolarity relative to the cLP and dLP solutions (Table 1).

## 4. Discussion

In this study, we show that the LP concentrations in the rat AH could be measured with a commercial LP ELISA kit, rather than having to use high-performance liquid chromatography (HPLC) with tandem mass spectrometry (MS), as reported recently for the analysis of AH LP concentrations in rabbits or dogs [21,22]. From these studies, LP concentrations in the AH were in the order of 10–200 ng/mL, with higher concentrations of LP in the rat AH relative to rabbits or dogs. It is possible that LP penetrated the AH better in rats as the rat cornea is thinner than that of rabbits, dogs, and humans [23,24,25]. Moreover, 15 μL eye drops were instilled onto the rat ocular surface here, whereas in previous studies, 30 μL (or one drop) was instilled onto the rabbit, dog, or human ocular surfaces [26,27]. As the rat ocular surface and tear volume are smaller, and the tear turnover rate is lower than in rabbits, dogs, and humans [23], it is possible that the LP concentration on the rat ocular surface was higher than in rabbits, dogs, and humans, and thus, more LP might have penetrated into the anterior chamber. Moreover, since the rat anterior chamber (14 μL) is smaller than that of rabbits (287 μL), dogs (770 μL), and humans (200–310 μL) [23,28,29,30,31], the LP concentration per unit volume in the rat anterior chamber in this study is likely to be higher than in rabbits, dogs, and humans.

When measured with standardized Schirmer strips, basal tear production was inhibited in patients subjected to topical anesthesia [32]. Since general and topical anesthesia were used for AH collection here, tear production might have decreased. Our data suggest that the clearance of eye drops from the rat ocular surface might be reduced; hence, there is the possibility that eye drops may persist longer at the ocular surface due to anesthesia than normal eyes.

Regarding the LPFA concentrations in the AH, there was a tendency for males to accumulate lower concentrations than females, and the significant differences in the concentration ratios in females were not evident in males. These eye drops may penetrate better in females, regardless of their composition, due to the larger body size of males of the same age, which could potentially produce differences in corneal thickness and eye sizes. LPFA concentrations in the AH were similar in the LPvHiHA and cLP rats, yet they tended to be higher than in the dLP rats despite the fact that the LP concentration in these dLP eye drops was the same as in the LPvHiHA eye drops (14 μg/mL), and less than in the cLP eye drops (50 μg/mL).

HA has viscoelastic and mucoadhesive properties [33,34], and it can be used as a vehicle for topical ophthalmic drug administration, alone or in combination with other compounds [35,36,37,38]. It can prolong the precorneal residence of pharmacologically active substances and increase their bioavailability [39,40,41]. Indeed, a formulation of LP with chitosan and HA was more shown to be effective in reducing the IOP in rabbits than LP alone [42]. The properties of HA depend on its concentration and molecular weight. The data obtained here suggest that the very-high-molecular-weight HA in the LPvHiHA was effective in keeping LP at the ocular surface, delaying its clearance relative to cLP. The higher AH concentrations obtained with LPvHiHA also suggest that this agent persisted sufficiently long at the ocular surface, and it was adequately transported across the ocular surface epithelium.

We noted that the ratio of the LPFA concentration in the AH was significantly lower for cLP/LPvHiHA and significantly higher for LPvHiHA/dLP relative to the LP concentration in the eye drops. A previous trial on humans reflected more effective IOP lowering by LPvHiHA containing 20 µg/mL LP than by commercial eye drops containing 50 µg/mL LP [43]. The combination of more effective transport of the active ingredient into the AH with less ocular surface inflammation may explain this observation [20]. Our data suggest that LPvHiHA produces better LP transport to the AH than cLP and dLP.

The LP concentration in AH may also be influenced by the osmolarity of the eye drops. Indeed, tear and AH osmolarity are generally about 300 mOSM, close to the osmolarity of serum. The transepithelial permeability of riboflavin solutions in cross-linking procedures are known to be affected by the NaCl concentration, which contributes to a shift in osmolarity [44]. Indeed, hypotonic drop osmolarity may promote the permeability of riboflavin through the epithelium. Here, the LPvHiHA eye drops (255 mOSM) were more hypotonic than the cLP (276 mOSM) and dLP (302 mOSM) solutions, and this may in part explain the enhanced performance of the LPvHiHA eye drops at the ocular surface.

The cLP eye drops contains a penetration enhancer, EDTA, which compromises the ocular surface when applied in the long term, and it disturbs the epithelium at the ocular surface [9]. No penetration enhancers were included in the LPvHiHA eye drops, suggesting that LPvHiHA can be effectively administered at the ocular surface with fewer adverse reactions. Corneal calcification may be produced by using phosphate-buffered eye drops [45,46], and one eye drop that caused calcification contained 50.9 mosm/L phosphate, 35 times the physiological value of 1.45 mosm/L. The phosphate concentration of LPvHiHA is within the physiological range (1.2 mmol/L), and thus, it is unlikely to provoke this type of side effect. Indeed, the European Authorities did not request the inclusion of a warning in the instructions for the use of Comfort Shield^®^ eye drops that were used to prepare LPvHiHA. Hence, these eye drops are unlikely to provoke long-term adverse effects such as those attributed to phosphates.

This study had several limitations. First, the comparisons between LPvHiHA and cLP eye drops may be affected by the differences in the LP concentrations used. The ocular pharmacokinetics of LP may not fall within a linear range at these concentrations and thus, further studies will be required to confirm the validity of these comparisons. Second, the number of animals/group was relatively small and the LPFA concentration in the AH was quite variable, meaning that further studies on larger numbers of animals will be necessary to analyze the detailed relationships between the LPFA concentrations in the AH of each group.

## 5. Conclusions

Although further studies on a larger number of rats are necessary, our initial data suggest that the LP concentration was maintained beyond 4 h in the anterior AH of rats instilled with LPvHiHA and close to the concentration achieved with cLP eye drops. The LP concentration in the AH of LPvHiHA animals tended to be higher than that associated with dLP, suggesting that very high MW HA may aid the transport of LP across the ocular surface epithelium without penetration enhancers.

## Figures and Tables

**Figure 1 pharmaceutics-16-00523-f001:**
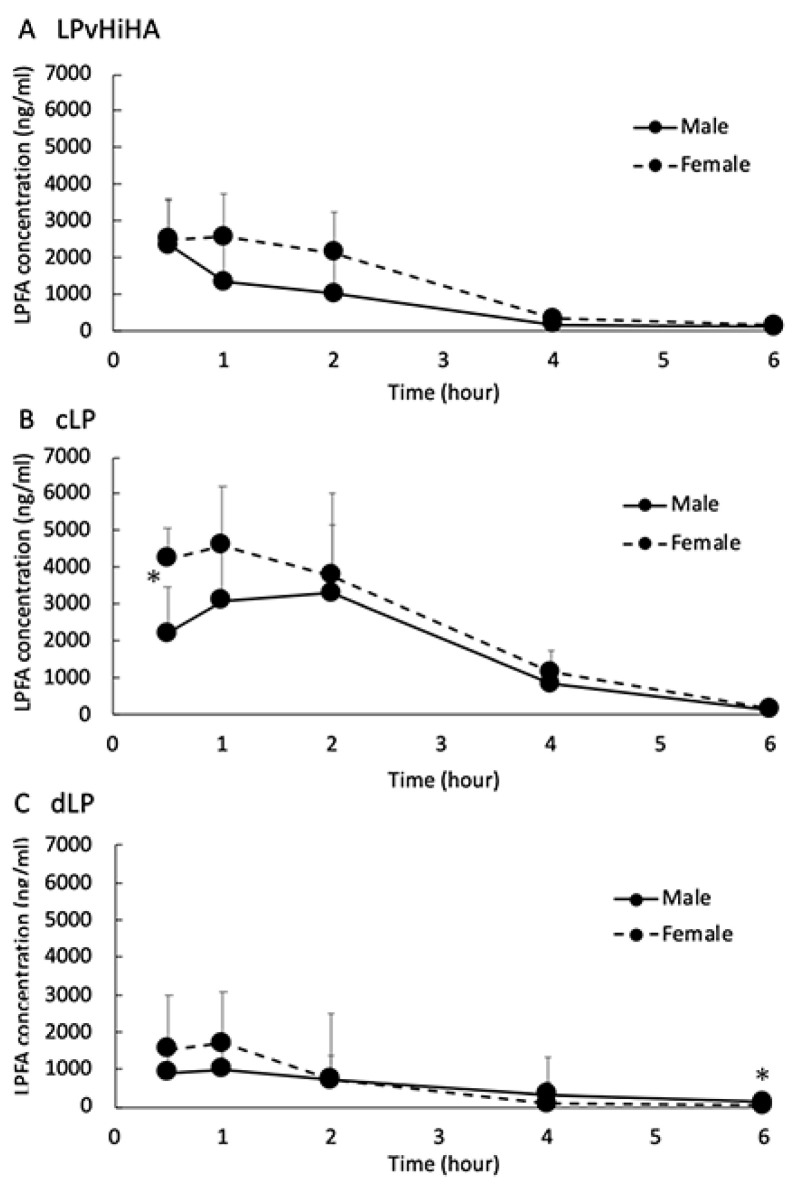
The time-dependent changes in LPFA concentration in the rat AH (males: black line, females: dotted line) after instilling LPvHiHA (**A**), cLP (**B**) or dLP (**C**). The *y*-axis is the LPFA concentration (ng/mL) and the *x*-axis is time (hour). The error bars represent the standard deviation, and the asterisks (*) indicate significant differences between male and female rats at each of the time points studied (*p* < 0.05, *n* = 5).

**Figure 2 pharmaceutics-16-00523-f002:**
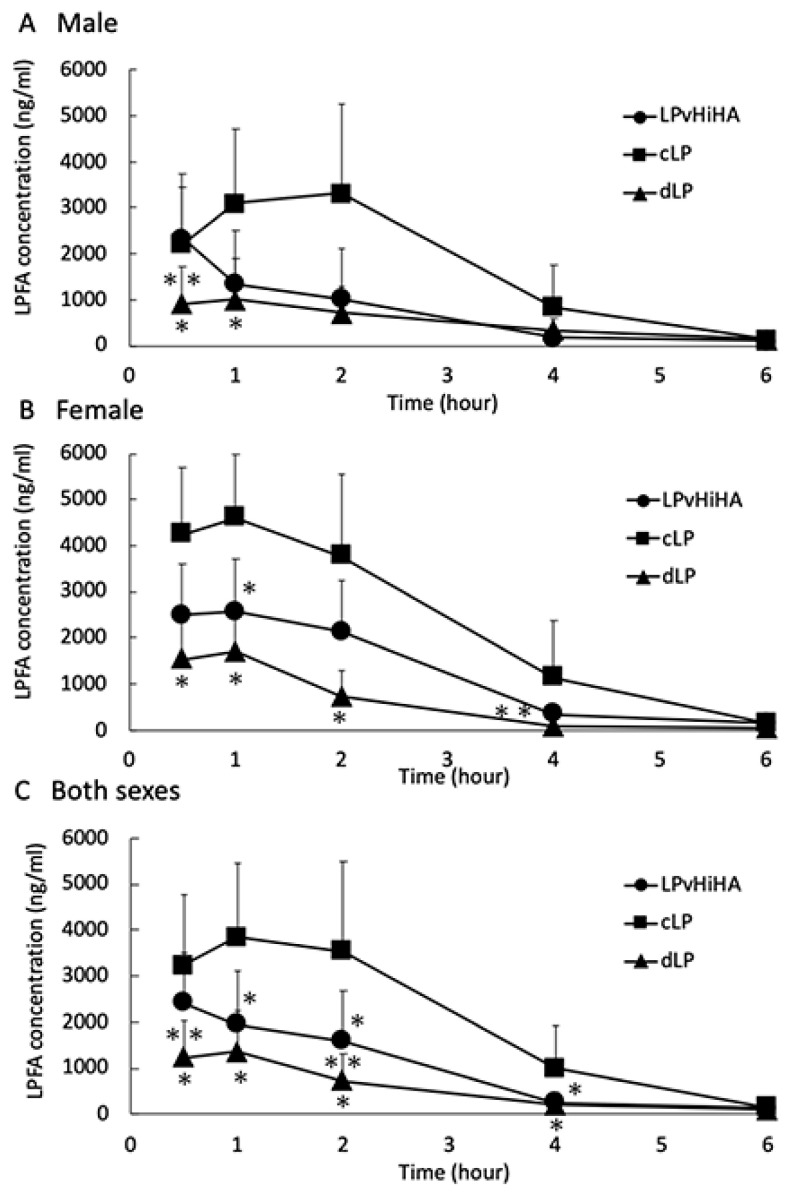
Comparison of the time-dependent changes in the AH LPFA concentration in males (**A**), females (**B**) or both sexes (**C**) of rats following instillation with LPvHiHA (round plots), cLP (square plots) or dLP (triangle plots). The *y*-axis shows the LPFA concentration (ng/mL) and the *x*-axis shows the time (hour). The error bars represent the standard deviations, a single asterisk (*) indicates significant differences between cLP and LPvHiHA or dLP at a given time point (*p* < 0.05, *n* = 5), and double asterisks (**) indicate significant differences between LPvHiHA and dLP at that time point (*p* < 0.05, *n* = 5).

**Figure 3 pharmaceutics-16-00523-f003:**
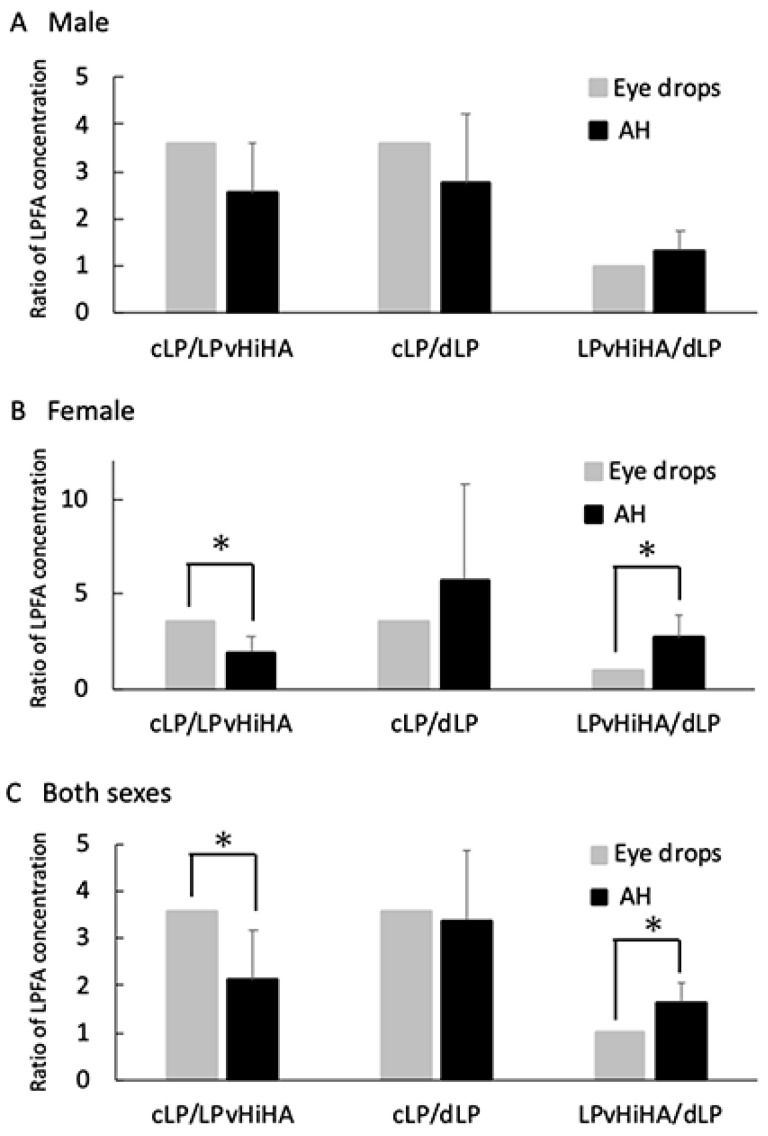
Comparison of the LP concentration ratios for each eye drop (gray bars) and the LPFA AH concentration ratios per group (black bar) in male (**A**) and female rats (**B**), and rats of both sexes (**C**). The values are the mean of 5 measurements from Figure 2 (0.5, 1, 2, 4, and 6 h) and the error bars represent the standard deviations. The asterisks (*) indicate significant differences between the eye drop concentration ratios and the LPFA concentration ratios in the AH (*p* < 0.05, *n* = 5).

**Table 1 pharmaceutics-16-00523-t001:** The ingredients, LP concentration and osmolarity of the LP eye drops.

LP Eye Drops	LPvHiHA	cLP	dLP
Ingredients	Very-high-molecular-weight hyaluronic acid	Boric acid	Boric acid
	Sodium chloride (NaCl)	Trometamol	Trometamol
	Disodium hydrogen phosphate (Na_2_HPO_4_)	Polyoxyethylene castor oil	Polyoxyethylene castor oil
	Sodium dihydrogen phosphate (NaH_2_PO_4_)	EDTA	EDTA
	Purified water (aqua purificate)	pH modifier	pH modifier
			Saline (0.9% NaCl)
LP concentrations	14 μg/mL	50 μg/mL	14 μg/mL
Osmolarity	255 ± 0.6 mOSM	276 ± 0.6 mOSM	302 ± 0.6 mOSM

**Table 2 pharmaceutics-16-00523-t002:** Latanoprost free acid (LPFA) concentration (ng/mL) in the aqueous humor (AH).

**Time (h)**	**Male**														
	**LPvHiHA**					**cLP**					**dLP**				
	**#1**	**#2**	**#3**	**#4**	**#5**	**#1**	**#2**	**#3**	**#4**	**#5**	**#1**	**#2**	**#3**	**#4**	**#5**
0.5	2706	2477	3242	2765	480	2778	1360	2100	1499	3282	1525	780	480	1654	130
1	2248	1666	1327	919	526	4356	2635	3125	668	4681	558	403	2232	1428	376
2	347	654	1575	1131	1376	2171	734	5907	2232	5489	76	59	1152	1575	685
4	110	195	298	85	167	1224	366	1537	180	886	931	227	199	112	164
6	85	75	121	114	101	95	112	219	95	125	157	151	248	57	69
**Time (h)**	**Female**														
	**LPvHiHA**					**cLP**					**dLP**				
	**#1**	**#2**	**#3**	**#4**	**#5**	**#1**	**#2**	**#3**	**#4**	**#5**	**#1**	**#2**	**#3**	**#4**	**#5**
0.5	2544	3549	831	1660	3865	3672	4245	2163	5872	5255	2063	681	2220	438	2261
1	1263	4033	3017	1121	3343	3672	3176	3999	6025	6129	2389	1353	2220	178	2324
2	1341	4282	1732	945	2396	2144	4282	2003	4103	6341	810	290	1169	56	1305
4	286	616	118	473	223	511	1065	606	3286	247	119	150	38	66	16
6	66	40	137	376	84	185	92	58	75	333	48	65	24	63	26

## Data Availability

The data presented in this study are available in this article (and Appendix A).

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
