# Peer review of "Therapeutic Aqueous Humor Concentrations of Latanoprost Attained in Rats by Administration in a Very-High-Molecular-Weight Hyaluronic Acid Eye Drop"

_pharmaceutics, 2024, doi:10.3390/pharmaceutics16040523_

Round 1

Reviewer 1 Report (Previous Reviewer 1)

Comments and Suggestions for Authors

This work examines the ocular bioavailability of latanoprost from three different solutions and investigates the impact of high molecular weight hyaluronic acid on drug absorption. The article is well-written, and the results and discussion are sound. However, some points should be addressed before publication:

Firstly, it is unclear why identical drug concentrations were not used in the three solutions. Without consistent drug concentrations, comparisons between the results are problematic, particularly given the uncertainty regarding the linearity of latanoprost ocular pharmacokinetics and whether the concentrations used fall within the linear range.

The article discusses the advantages of EDTA elimination, but it fails to address the inclusion of phosphates in the hyaluronan formulation. The potential long-term adverse effects of phosphates, such as soft keratopathy due to hydroxyapatite crystal deposition in the superficial stroma and Bowman's membrane, should be acknowledged and discussed by the authors.

Lastly, the captions of the graphs should specify whether the deviation represents the standard error or the standard deviation. This clarification is essential for accurately interpreting the data presented.

Author Response

This work examines the ocular bioavailability of latanoprost from three different solutions and investigates the impact of high molecular weight hyaluronic acid on drug absorption. The article is well-written, and the results and discussion are sound. However, some points should be addressed before publication:

Firstly, it is unclear why identical drug concentrations were not used in the three solutions. Without consistent drug concentrations, comparisons between the results are problematic, particularly given the uncertainty regarding the linearity of latanoprost ocular pharmacokinetics and whether the concentrations used fall within the linear range.

A previous trial on a human subject had resulted in more effective IOP lowering by LPvHiHA containing 20 µg/ml latanoprost as compared to commercial latanoprost eye drops containing 50 µg/ml [44]. LPvHiHA eye drops used in this study have a lower concentration of 14 µg/ml, we diluted the commercial latanoprost eye drops (cLP) to match the concentration (14 µg/ml ) and used them in this study. As reviewer 1 indicated, comparisons between LPvHiHA (14 µg/ml ) and cLP (50 µg/ml ) eye drops are given the uncertainty regarding the linearity of latanoprost ocular pharmacokinetics and whether the concentrations used fall within the linear range because latanoprost concentrations in eye drops were different. We had added the following sentences in the discussion as limitation;

This study had several limitations. First, the comparisons between LPvHiHA and cLP eye drops may be affected by the differences in the LP concentrations used. The ocular pharmacokinetics of LP may not fall within a linear range at these concentrations and thus, further studies will be required to confirm the validity of these comparisons.

The article discusses the advantages of EDTA elimination, but it fails to address the inclusion of phosphates in the hyaluronan formulation. The potential long-term adverse effects of phosphates, such as soft keratopathy due to hydroxyapatite crystal deposition in the superficial stroma and Bowman's membrane, should be acknowledged and discussed by the authors.

Thank you very much for your valuable comments. We had added the following sentences in the discussion;

Corneal calcification may be produced by using phosphate buffered eye drops [45,46] and one eye drop that caused calcification contained 50.9 mosm/L phosphate, 35 times the physiological value of 1.45 mosm/L. The phosphate concentration of LPvHiHA is within the physiological range (1.2 mmol/L) and thus, it is unlikely to provoke this type of side effect. Indeed, the European Authorities did not request the inclusion of a warning in the instructions for the use of Comfort Shield® eye drops that were those used to prepare LPvHiHA. Hence, these eye drops are unlikely to provoke long-term adverse effects such as those attributed to phosphates.

Lastly, the captions of the graphs should specify whether the deviation represents the standard error or the standard deviation. This clarification is essential for accurately interpreting the data presented.

As reviewer 1 indicated, we had added the following sentence in the figure legends;The error bars represent standard deviations.

Reviewer 2 Report (Previous Reviewer 4)

Comments and Suggestions for Authors

The authors have addressed my concerns appropriately. The paper is ready for publication. 

Comments on the Quality of English Language

The English Language is adequate. 

Author Response

The authors have addressed my concerns appropriately. The paper is ready for publication. Comments on the Quality of English Language. The English Language is adequate. 

Thank you very much for your review.

Reviewer 3 Report (Previous Reviewer 5)

Comments and Suggestions for Authors

The authors provided with a revised manuscript of a previous one sent to Pharma. in 2023.

One of the main concerns is the low number of animals. The high variability of LPFA in AH (range 480-3242) would prevent a high power statistic on 5 animals/group (or 10 if we add both sexes).

However, the tendency observed is valuable, HA could be used as a carrier.

Please add all these limitations to the Discussion

Line 54 - A previous research by us...   / Please reformulate

Author Response

The authors provided with a revised manuscript of a previous one sent to Pharma. in 2023.

One of the main concerns is the low number of animals. The high variability of LPFA in AH (range 480-3242) would prevent a high power statistic on 5 animals/group (or 10 if we add both sexes).

However, the tendency observed is valuable, HA could be used as a carrier.

Please add all these limitations to the Discussion

As reviewer 2 indicated, the number of animals/group was small, and there is the high variability of LPFA concentration in AH. We had added the following sentences in the discussion as limitation;

Second, the number of animals/group was relatively small and the LPFA concentration in the AH was quite variable, such that further studies on larger numbers of animals will be necessary to analyze the detailed relationships between the LPFA concentrations in the AH of each group. 

Line 54 - A previous research by us...   / Please reformulate

We had reformulated the sentences of Line 54- as a reviewer 2 as follows;

From:

A previous research by us found that the combination eye drop (latanoprost with vHiHA: LPvHiHA) had favorable effects on ocular surface staining and tear functions [21]. Our recent research data suggested that wild type mice receiving LPvHiHA eye drops had a better ocular surface health status compared to the mice receiving solitary commercial LP eye drops at 7 days as evidenced by significantly lower corneal vital staining scores, higher ZO-1 mRNA expressions and higher conjunctival goblet cell densities [21]. In our previous work, the LPvHiHA mouse group also had significantly lower ocular surface-tear film unit inflammation as evidenced by significantly lower conjunctival CD45+ staining inflammatory cell counts and lower tear IL-6 as well as low IL-1b levels [21]. Whether these favorable effects were due to a better penetration into the aqueous humor (AH) remained to be investigated.

To:

We previously demonstrated that delivery of LP in an eye drop with vHiHA (LPvHiHA) had favorable effects on ocular surface staining and tear functions [21]. Indeed, more recent data indicated that the ocular surface of wild type (WT) mice receiving LPvHiHA eye drops over 7 days was in better health than that of mice receiving a commercial LP (cLP) eye drop. This improvement was witnessed through the significantly lower corneal vital staining scores, stronger ZO-1 mRNA expression and higher conjunctival goblet cell densities [21]. In fact, the mice administered LPvHiHA also had significantly less ocular surface-tear film inflammation, as evident through the significantly fewer conjunctival inflammatory CD45+ stained cells and lower tear IL-6, as well as lower IL-1b levels [21]. However, whether these favorable effects were due to better penetration into the aqueous humor (AH) remained to be assessed.

Round 2

Reviewer 1 Report (Previous Reviewer 1)

Comments and Suggestions for Authors

After evaluating the authors' responses, I consider that the article can be published in pharmaceutics.

This manuscript is a resubmission of an earlier submission. The following is a list of the peer review reports and author responses from that submission.

Round 1

Reviewer 1 Report

Comments and Suggestions for Authors

Eye drop instillation and collection of Aqueous Humor (AH): Are the animals completely anesthetized during the entire trial? If so, how does this affect the clearance of the drug from the ocular surface? Has this been taken into account? these aspects have to be included in the manuscript

The sentence “The dilutions of eye drops in each group were as follows; LPCS (group1, 14 mg/ml,i.com medical GmbH, Munich, Germany), cLP (group2, 50 mg/ml, Rhoto Nitten Co., Ltd., Nagoya, Japan), and dLP (group3, 50mg/ml cLP were diluted to 14 mg/ml with saline). 15ml of eye drops were instilled in each group (n=5)” is unclear. The Authors should include a separate section on the preparation of ocular dissolutions. In the case of LPCS, is it a formulation prepared by i.com medical, or is just a dissolution of LP in  Comfort Tears®?

Figure 3 is speculative since for LPCS there is only one concentration point. Eliminate it. I suggest the authors replace it with a table including non-compartmental (or compartmental if you prefer) pharmacokinetic parameters, i.e. Cmax, tmax, ke elimination counter, area under the curve (AUC), and mean residence time (MRT).

Reviewer 2 Report

Comments and Suggestions for Authors

I’ve read with great interest the manuscript entitled “The first evidence in attaining therapeutic aqueous humor concentrations by a novel latanoprost and very high molecular weight hyaluronic acid eye drop (Comfort Shield) in rats.”. I recognize the effort made by the authors but I think the manuscript is unacceptable for publication in its current form. I hope that my response does not discourage the authors. I’m providing them with my point of view on certain topics with the sincere aim to help them to improve the manuscript for future submissions.

·         General comments

The manuscript is aimed to describe the potential use of very high weight hyaluronic acid as a vehicle to improve the penetration of latanoprost to the anterior chamber of the rat eye, using lower concentrations of the active compound -in comparison with other commercial formulations-. Although the rationale of the treatment groups seems appropriate, I miss the sample size estimation and/or the statistical power of the data comparison. Given the high dispersion of some of the values presented (figures 1 and 2, table 1), I consider that more observations should be included. In addition, since only female animals were used, I think that also male animals should be included, and a potential sex dimorphism on the effects, excluded. As a quality control, the aqueous humor of the contralateral, non-treated eye should be analyzed, at least in some pilot studies. Moreover, the treatment received by each animal should be masked for the experimenters.

I’m also confused with the data reported by the authors, since they report values in the range ≈1000 – 6000 ng·ml-1 during the first hour after the topical administration of latanoprost, and describe that had to dilute the samples by 5000 times to be in the range of the ELISA standard curve (that ranged among 3.9 and 500 pg·ml-1). Previous works in both rabbits (Zhou et al, 2020; PMID: 32310714) and humans (Sekine et al, 2018; PMID: 29323612), have reported aqueous humor concentrations after topically applied latanoprost in the range of 10 ng·ml-1 (human) to 70 ng·ml-1 (rabbit). I think that the disparity of the values reported by the authors when compared to previously published ones needs further analysis and discussion.

In summary, I consider that further experiments and a better discussion of the results are needed.

·         Specific comments.

1.       Avoid to use the commercial trademark of the compound tested (Comfort Shield), specially at the title. Only if it is strictly necessary to use it along the manuscript, please indicate that is a trademark. Otherwise, use another denomination.

2.       Avoid to duplicate data presentation: table 1 and figure 1 are depicting exactly the same data. I would suggest to remove current table 1 from the manuscript and to submit it as supplementary table.

3.       Current figure 2 could be merged to current figure 1, as a new panel D. Therefore, both the mean values and the individual values could be presented at a single place. If there is a normative value of therapeutic LPFA concentration (or a range of values), please highlight it as horizontal reference lines at the graphs. Therefore, it would be much easier for the readers to see if the experimental values are within the normative range of values.

4.       Current table 2 should be moved to the methods section, since it summarizes the characteristics and composition of the solutions.

5.       The experiment whose results are shown in figure 3 is not described at the methods section. Neither the number of observations, nor the timepoint at which the measurements were taken are indicated along the manuscript.

6.       Please avoid to use the reiteration of the group number and the corresponding treatment (for example, Group1 and its treatment, LPCS). Choose only one denomination and/or acronym for each group and use solely them along the manuscript.

7.       Since there are more than three groups, and the observations within each subject are repeated several times, the appropriate statistical test to be used is a repeated measurements ANOVA, rather than a non-paired Student’s t test.

Reviewer 3 Report

Comments and Suggestions for Authors

I reviewed the manuscript "The first evidence in attaining therapeutic aqueous humor concentrations by a novel latanoprost and very high molecular weight hyaluronic acid eye drop (Comfort Shield) in rats" by Higa et al.

First of all, authors should change the manuscript type at the top of the first page, as this manuscript is not an article but a brief report.

Unfortunately, I am very critical towards this manuscript. In fact, the experimental design and the presentation of the data are confusing:

- Figure 1 is completely unnecessary and also confusing

- Figure 2 is fine, but I don't understand the significance of testing the concentrated cLP, as only preparations with the same drug concentration can be correctly compared, or at leas they shoudl be normalized. Also I don't understand why the Y axis is 0-7000 ng/mL while in the methods authors write that they diluted samples to be in a range of 4-500 pg/mL (different orders of magnitude). Furthermore, the SD reported is very high, are you sure that at 1h the two values are significantly different? I tried to calculate statistics with Student t test using the numbers that authors put in the first table, I get a p = 0.3, not significant. 

- Table 2: you should report the SD on the measurement of Osmolarity

Furthermore, no information is given regarding the preparation of LP with vHMWHA, which should be the core of this article. Did authors prepare this eye drop or is it commercial? In any case authors should report information about the concentration of HA (which they correctly state in the discussion that could be a critical parameter to assess its impact on the clearance of the eyedrops), rheological properties of the preparation compared to the commercial, storage stability, mucoadhesion etc. 

Without all this information and with these poor results, I don't think this report is significant for publication, as the only interesting result is given by a slight increase in the LP concentration when using the HA compared to the diluted cLP. 

I think authors should do more experiments with an in-depth investigation of the mechanisms and prepare a full research article.

Reviewer 4 Report

Comments and Suggestions for Authors

In this work, the time-wise changes of novel Latanoprost (LP), with high molecular weight hyaluronic acid (vHMWHA) in the aqueous humor in rats were evaluated using ELISA. One of the good thing about this formulation is the absence of EDTA which is known to  cause adverse effects to the ocular surface.

1.       Despite the exciting results, this might be too small for an “article”, a “communication” or “short letter” would be better. I will leave this to the editors to decided.

2.       I don’t think it’s necessary to include the detail of your study in the abstract, such as the number of animals in each treatment group, nor how much of the eye-drop given.

3.       Line 42: Give full name of EDTA before using abbreviation

4.       Put a number for each sub-headings, For example/ 2.1 Animal

Reviewer 5 Report

Comments and Suggestions for Authors

The article presents a further investigation of the rather new DDS, involving HA formulations. The paper proves the posibility of determening LP in aqueous humor, with some limitations.

line 16- Wistar rats (are there Wister rats?). Please change througout the entire document

Line 89-90 delete second the

Statistical test should be reconsidered. The number of samples are too low. Are they normally distributed?

Also, there is data that seems wrong - for insance #3 in cLP is 3999ng, sugesting dramatic increase in LPA at 6 hours, comparing to 606ng at 4 hours. Because of this (and other data), figure 2 may show skewed statistics. e.g for LPCS, is it #3 and 4 is true, or #2 and 5? SD is too high! The power beta for the statistics should be obtained with a much higher number of samples. It is likely that LPCS showes better results compared to diluted samples, but this is expected, due to persistance on ocular surface. Only one eye was evaluated?

Figure 3 - in discussion, is it based on present study? The present study does not investigate different concentrations. A liniar relationship LP in eye drops - LPFA in aqueous humor is not necessary true. The clinical effect is proven not to be a liniar corelation to concentration (e.g twice the commercial dose, less effect).

In conclusion, the main concerns of the reviewer are the low number of animals (5 per lot), and the predictibility of ELISA results (to much difference, concerns about precision; this could be also alleviated with higher number of animals).